# Effect of a freeze-dried coffee solution in a high-fat diet-induced obesity model in rats: Impact on inflammatory response, lipid profile, and gut microbiota

**Marilia Hermes Cavalcanti**[1☯¤a]*, **João Paulo Santos Roseira**[2☯¤b], **Eliana dos Santos Leandro**[1,3☯¤a], **Sandra Fernandes Arruda**[1,3☯¤a]

1 Postgraduate Program in Human Nutrition, Faculty of Health Sciences, Campus Universitário Darcy Ribeiro, Universidade de Brasília, Brasília, Distrito Federal, Brazil, 2 Department of Animal Science, Universidade Federal de Viçosa, Viçosa, Minas Gerais, Brazil, 3 Department of Nutrition, Faculty of Health Sciences, Campus Universitário Darcy Ribeiro, Universidade de Brasília, Brasília, Distrito Federal, Brazil

☯ These authors contributed equally to this work.
¤a Current address: Postgraduate Program in Human Nutrition, Faculty of Health Sciences, Campus Universitário Darcy Ribeiro, Universidade de Brasília, Brasília, Distrito Federal, Brazil
¤b Current address: Department of Animal Science, Universidade Federal de Viçosa, Viçosa, Minas Gerais, Brazil
* mariliaunb@outlook.com

**Data Availability Statement:** All relevant data are within the manuscript and its Supporting Information files.

## Abstract

Coffee beans contain high polyphenol content, which have the potential to modulate the intestinal microbiota, and possibly attenuate weight gain and the associated dyslipidemia. This study investigated the effect of freeze-dried coffee solution (FCS) consumption on physiological parameters, lipid profile, and microbiota of Wistar rats fed a high-fat diet (HF) or control diet (CT). FCS combined with a high-fat diet increased the fecal and cecal *Bifidobacterium* spp. population and decreased the cecal *Escherichia coli* population and intestinal Il1b mRNA level. Regardless of the diet type, FCS increased the serum high-density lipoprotein cholesterol (HDL-C); however, it did not affect body weight, food intake, low-density lipoprotein, triglycerides, fecal bile acids, and intestinal Il6 mRNA levels. The high-fat diet increased weight gain, hepatic cholesterol and triglycerides, fecal bile acids, and the fecal and cecal *Lactobacillus* spp. population, and reduced food intake, the fecal *E. coli* population, and intestinal Il6 mRNA level. The results suggest that FCS consumption exhibits positive health effects in rats fed a high-fat diet by increasing *Bifidobacterium* spp. population and HDL-C reverse cholesterol transport, and by reducing Il1b mRNA level. However, FCS administration at a dose of 0.39 g/100 g diet over an eight-week period was not effective in controlling food intake, and consequently, preventing weight gain in rats of high-fat diet-induced obesity model.

**Funding:** The author Marilia Hermes Cavalcanti received financial scholarship for the Coordenação de Aperfeiçoamento de Pessoal de Nível Superior (CAPES) (code 001) and financial aid for the research by Fundação de Apoio a Pesquisa do Distrito Federal (FAP-DF). The funders had no role in study design, data collection and analysis, decision to publish, or preparation of the manuscript.

**Competing interests:** The authors have declared that no competing interests exist.

## Introduction

Coffee beverages are consumed worldwide. The United States and Brazil are the largest consumers of coffee, accounting for 28% of the total consumption of green coffee beans in the world. Brazilians consume an average of 5.8 kg of coffee per year, whereas the average global coffee consumption is 1.3 kg/person [1]. *Coffea arabica* L. and *Coffea canephora Pierre* are commercially significant species of coffee in the world; *C. arabica* alone represents 65% of the global coffee production [2].

Many potential health benefits have been associated with regular coffee consumption. A meta-analysis study suggested that regular coffee consumption of 0.5–5 cups/day was associated with a reduced risk of metabolic syndrome (MetS) [3]. Data obtained from the 2012–2015 Korea National Health and Nutrition Examination Survey also showed that moderate coffee consumption (3–4 times/day) was inversely associated with MetS in adults; however, no association was observed with heavy coffee consumption ($\geq$ 5 times/day) [4]. Adult male Wistar rats fed a high-carbohydrate and high-fat diet for 16 weeks supplemented with 5% spent coffee grounds during the last eight weeks showed lower body weight, plasma triglycerides, and non-esterified fatty acids than in those not fed coffee [5]. A 24-week placebo control trial conducted in overweight and insulin-resistant individuals found that consuming four cups/day of coffee did not affect dyslipidemia biomarkers, although it promoted moderate loss of fatty mass [6].

Considering high content of polyphenols (chlorogenic acids, alkaloids, polyphenols, caffeine, and trigonelline) in coffee beans, many health benefits associated with coffee consumption have been attributed to these compounds. The complex structure of dietary polyphenols makes them a substrate for intestinal microbiota, as 95% of these compounds reach the large intestine and therefore interact with the gut microbiome [7]. A recent review study suggested that polyphenols act as prebiotics, inhibiting the growth of pathogenic bacteria without affecting or stimulating beneficial bacteria [8]. Cowan *et al.* [9] observed protective effect of coffee on the gut microbiota of rats fed a high-fat diet, where coffee attenuated the typical increase in the Firmicutes/Bacteroidetes ratio observed in obesity models.

Since early studies have observed that germ-free mice gained less weight and fat mass compared to those harboring a gut microbiota when fed a high-fat diet, besides being resistant to developing obesity when fed a high-fat diet, the gut microbiota has been indicated as a potential environmental factor in the etiology of obesity and its comorbidities such as dyslipidemia [10, 11]. Compared to lean mice, the *ob/ob* animal obesity model presented a reduction in Bacteroidetes and a proportional increase in Firmicutes [12]. A similar profile was observed in individuals with obesity [13], with the presence of fewer numbers of *Bacteroides*, representing the most abundant genus in the Bacteroidetes phylum in humans [14], than that in non-obese individuals [15]. Moreover, low counts of some *Bifidobacterium* and *Enterococcus* species were associated with weight gain and dyslipidemia [13, 16]. Some *Lactobacillus* species were associated with weight gain, whereas others such as *Lactobacillus paracasei* and *Lactobacillus plantarum* were considered probiotic strains that promoted weight loss [13, 16, 17].

The metabolism of bile acids by gut bacteria also seems to mediate the relationship between weight gain and serum lipid profile of the host and the gut microbiome composition [18]. An increase in the activity of bacterial bile salt hydrolase enzymes, which deconjugate glycine-and taurine-conjugated bile acids to generate unconjugated bile acids, was associated with a reduction in weight gain, serum cholesterol, and liver triglycerides in mice with the typical microbiota. Therefore, it has been suggested the gut microbiota modifies the host's lipid metabolism, and consequently, improves obesity and metabolic syndrome through bacterial bile salt hydrolase enzymes [19].

Most studies evaluating the benefits of coffee in rats fed a high-fat diet used coffee extracts, instant coffee, or purified bioactive components of coffee. To date, there are no studies available that have evaluated the effect of incorporating coffee into food with the aim of preventing obesity in rats. Freeze drying has been considered the best method for preserving the chemical properties of foods. Freeze-dried coffee solution is a new product that can be incorporated into different foods. Therefore, this study was aimed to investigate the effect of a freeze-dried coffee solution mixed with a high-fat diet on physiological parameters, lipid profile, and microbiota of rats in a high-fat diet-induced obesity model.

## Materials and methods

### Preparation of freeze-dried coffee solution

A 10% ground coffee solution, consisting of a blend of *C. arabica* grains (Ponto Aralto, Jundiaí, SP, Brazil), was prepared by filtration using 100% cellulose filter paper (Original, no. 103, *Melitta*®, São Paulo, SP, Brazil) and water at 90˚C. The coffee solution was freeze-dried using an industrial freeze dryer (Beta 2–8 LSC PLUS Martin Christ, Nova Analítica Ltda, São Paulo, SP, Brazil) and stored at −80˚C until further use. The amount of coffee solution added to the diet of the rats was estimated based on the Brazilian population's average consumption of 163 mL of coffee [20] and 1,290 g of food per day. Considering adult rats consume an average of 25 g of food per day, the equivalent average coffee intake was estimated to be 3.15 mL/day, corresponding to 126 mL coffee/kg diet. After freeze-drying process, 126 mL of 10% coffee solution yielded 3.9 g of powder. The freeze-dried coffee solution (FCS) was added to the diet at a proportion of 3.9 g/kg of diet. Previous characterization of a commercial coffee brand of *C. arabica* species (Ponto Aralto, Jundiaí, SP, Brazil) found 19.93, 4.22, 5.14, 9.63 and 18.99 mg of caffeine, 3-, 4-, 5-, and total caffeoylquinic acid/g, respectively [21].

### Ethics statement

The experimental protocol was approved by the Animal Care and Use Committee of the University of Brasília (protocol no. 25/2018, approved on 05/08/2018) in accordance with the Brazilian National Council for Animal Experimentation Control (CONCEA) and the Guide for the Care and Use of Laboratory Animals [22].

### Animals

Twenty-eight male Wistar rats (Institute of Biomedical Sciences, University of São Paulo, SP, Brazil), 21-days old with an average body weight of 67.37 ± 6.04 g, were housed individually in stainless steel cages in a room under 12/12 h light/dark cycle at 22 ± 1˚C. The diet was provided from 12 pm to 8 am, and rats had free access to water.

The animals were fed the AIN-93G control diet [23] for seven days for acclimatization; the rats were then assigned to the following experimental groups (seven rats/group) and fed the corresponding diets for a period of 56 days: control group (**CT**-): AIN-93G diet; high-fat group (**HF**-): AIN-93G diet with 58% of fat; coffee group (**CF**+): AIN-93G diet with 3.9 g of FCS/kg of diet; and high-fat + coffee group (**HF**+): AIN-93G diet with 58% of fat (51.9% of lard and 6.1% of soy oil) and 3.9 g of FCS/kg of diet. The fat percentage was determined according to the Research Diets, Inc. diet-induced obesity model (D12492, Research Diets, Inc., New Brunswick, NJ, USA). Food intake was recorded daily, and the body weight was recorded weekly.

During the experimental period, the feces of each animal were collected daily, pooled in the same tube for a week, and stored at −80˚C. This provided eight-week fecal samples from each

animal at the end of the experimental period. These samples were analyzed at the first and eighth week of the treatment.

At the end of the treatment period, after a 7 h fasting period, the animals were anesthetized under 3% isoflurane (BioChimico, Rio de Janeiro, RJ, Brazil) and euthanized by exsanguination *via* cardiac puncture. The large intestine (from the ileocecal valve to the rectum) and cecum were excised. The remaining fecal content in the large intestine and cecum content were collected in sterile tubes and stored at −80˚ C. Subsequently, the large intestine was washed with 0.9% saline solution at 4˚C, immediately frozen in liquid nitrogen, and stored at −80˚C until further analysis.

## Lipid profile

Total cholesterol (CLT), high-density lipoprotein (HDL) cholesterol, low-density lipoprotein (LDL) cholesterol, and triglyceride (TG) concentrations were measured in the serum and liver using commercial enzymatic/colorimetric assay kits (BioClin, Belo Horizonte, MG, Brazil), according to the manufacturer's protocol. Total lipid extraction from the liver was performed as described by Vieira *et al*. [24]. Briefly, 50 mg of the liver was homogenized in 1 mL of iso-propanol and centrifuged at 2000 g for 10 min at 4˚C. The supernatant was removed and stored at −80˚C until further analysis.

## Bile acid concentration in feces

Aliquots from the feces samples of the first and eighth week of the treatment was freeze-dried at −45˚C for 48 h. The freeze-dried samples were macerated in a porcelain mortar using liquid nitrogen. Total bile acid was extracted in ethanol as described by Kanamoto *et al*. [25] and Tamura *et al*. [26], and the total bile acid content was measured by fluorometry using a commercial assay kit (Sigma-Aldrich, St. Louis, MO, USA), following the manufacturer's instructions.

## Determination of fecal microbiota composition

**Extraction of DNA from fecal sample and cecum content.**   The DNA from the fecal samples was extracted using QIAamp PowerFecal DNA kit (Qiagen, Hilden, RP, Germany), according to the manufacturer's protocol, with the following modifications: 150 mg of the fecal sample was used and the samples were homogenized using a cell/tissue disruptor (L-beader 6, Cotia, São Paulo, SP, Brazil) following a schedule of 2 cycles of 2500 rpm for 15 s for cecum content and 3 cycles of 2500 rpm for 15 s for fecal sample. The samples were placed on ice for 30 s between each cycle.

The DNA samples were quantified by determining the absorbance at 260 nm using the equation $A_{260 \text{ nm}} \times 50 \times$ dilution factor, whereas DNA purity was assessed by determining the absorbance ratios $A_{260 \text{ nm}}/A_{280 \text{ nm}}$ (approximately 1.8–2.0) and $A_{260 \text{ nm}}/A_{230 \text{ nm}}$ (approximately 2.0) [27] and quality of DNA was evaluated by agarose gel electrophoresis.

**Real-time PCR analysis.**   The gut microbiota composition of *Bifidobacterium* spp., *Lactobacillus* spp., *Escherichia coli*, *Bacteroides* spp., and *Enterococcus* was evaluated using quantitative real-time polymerase chain reaction (qPCR, StepOnePlus System, Applied Biosystems, Foster City, CA, USA). Every analysis was performed in triplicate using 5 μL of Fast SYBR Green Master Mix (Applied Biosystems, Foster City, CA, USA), 2 μL of sample DNA or standard, 0.2 μL of sense and antisense oligonucleotides (100 nM; Table 1), and water to a final assay volume of 10 μL. The initial DNA denaturation occurred at 95˚C for 20 s, followed by 40 cycles of denaturation at 95˚C for 3 s, annealing of oligonucleotides, and extension at 59–60˚C for 30 s.

**Table 1. Primers and reaction conditions for the bacterial genera analyzed by RT-qPCR.**

| Microorganisms | Primer sequence (5'- 3') | T (ºC) | Reference |
|---|---|---|---|
| *E. coli* | CATGCCGCGTGTATGAAGAA (F) | 59 | [28] |
| | CGGGTAACGTCAATGAGCAAA (R) | | |
| *Enterococcus* spp. | CCCTTATTGTTAGTTGCCATCATT (F) | 60 | [29] |
| | ACTCGTTGTACTTCCCATTGT (R) | | |
| *Bifidobacterium* spp. | AGGGTTCGATTCTGGCTCAG (F) | 60 | [30] |
| | CATCCGGCATTACCACCC (R) | | |
| *Lactobacillus* spp. | TGGATGCCTTGGCACTAGGA (F) | 60 | [31] |
| | AAATCTCCGGATCAAAGCTTACTTAT (R) | | |
| *Bacteroides* spp. | GAGAGGAAGGTCCCCCAC (F) | 60 | [32] |
| | CGCTACTTGGCTGGTTCAG (R) | | |

F: forward primer; R: reverse primer; T: annealing temperature.

Standard curves were constructed for each experiment using five sequential dilutions of bacterial genomic DNA from pure cultures ranging from 200 ng to 0.064 ng. The results were expressed as log10 of 16S rRNA copy number per gram of feces as described by Talarico *et al.* [33]. The different strains used were obtained from the American Type Culture Collection (ATCC) (*E. coli* ATCC 25992; *E. faecalis* ATCC 19433; *Bacteroides* ATCC 25285), commercial culture (*Bifidobacterium* spp. BL 04), and the Tropical Cultures Collection (*L. plantarum* UnB SBR64.1 MK5114407).

## Determination of mRNA levels of pro-inflammatory genes

**RNA extraction and cDNA synthesis.** Total RNA was extracted from the large intestine using TRIzol reagent (Invitrogen Inc., Burlington, ON, Canada), according to the manufacturer's protocol. Total RNA sample was quantified by measuring the absorbance at 260 nm ($A_{260\ nm} \times 40 \times$ dilution factor), and its purity was evaluated by the absorbance ratios $A_{260\ nm}/A_{280\ nm}$ and $A_{260\ nm}/A_{230\ nm}$ [27]. The integrity of the RNA bands was verified by agarose gel electrophoresis.

cDNA synthesis was performed using High-Capacity cDNA Reverse Transcription Kit with RNase Inhibitor (Applied Biosystems, Foster City, CA, USA).

**Quantification of transcriptional levels of pro-inflammatory genes Il1b, Il6, and Tnfa.** The transcript levels of interleukin 1 beta (Il1b), interleukin 6 (Il6), and tumor necrosis factor alpha (Tnfa) in the large intestine were determined by RT-qPCR. The reaction was performed using 2.0 μL of cDNA (final concentration of 20 ng), 5.0 μL of Fast SYBR Green Master Mix (Applied Biosystems, Foster City, CA), and 0.2 μM/L (final concentration) of each primer (Table 2), to a final volume of 10 μL. The RT-qPCR reactions were performed at 95˚C for 20 s followed by 40 cycles of 95˚C for 3 s and 60˚C for 30 s. The primers used are shown in Table 2. All samples were assayed in triplicate, normalized to the housekeeping gene β-actin, and the amplification specificity of each amplicon was analyzed using the dissociation curve. The relative quantification of each target gene mRNA level was determined using the 2−ΔΔCT method [34].

## Statistical analysis

The data obtained from microbial populations and bile acids were analyzed according to a completely randomized 2 × 2 × 2 factorial design, considering two diets (CT and HF) with (+) or without (−) the addition of coffee and two treatment periods (the first and eighth week).

The variables from the other experiments were analyzed according to a completely randomized 2 × 2 factorial design (diet and coffee). Homogeneity of the variances between treatments

**Table 2. Sequences of primers used for RT-qPCR assay of Il1b, Il6, Tnfa and Actb.**

| Gene | Primer sequence (5'- 3') | Reference |
|---|---|---|
| Interleukin 1 beta (Il1b) | CACCTCTCAAGCAGAGCACAG (F) | [35] |
| | GGGTTCCATGGTGAAGTCAAC (R) | |
| Interleukin 6 (Il6) | GCCAGAGTCATTCAGAGCAATA (F) | [36] |
| | GTTGGATGGTCTTGGTCCTTAG (R) | |
| Tumor necrosis factor alpha (Tnfa) | AAATGGGCTCCCTCTCATCAGTTC (F) | [35] |
| | GTCGTAGCAAACCACCAAGCAGA (R) | |
| β Actin (Actb) | GTCGTACCACTGGCATTGTG (F) | [37] |
| | CTCTCAGCTGTGGTGGTGAA (R) | |

F: forward primer; R: reverse primer.

was assumed, and after analysis of variance, significant interactions between the factors were unfolded and compared using the F test and Tukey test. The box plot method was used to remove outliers, and the initial weights of animals were used as covariates. A critical probability level of 0.05 was adopted for type I errors using PROC MIXED from SAS version 9.4 (SAS Institute Inc., Cary, NC, USA).

## Results

### Effect of freeze-dried coffee solution consumption on food intake and body weight

The effect of FCS on food intake and body weight gain was examined in rats fed the control or a high-fat diet (Table 3). The addition of FCS to the control (CT+) or a high-fat diet (HF+) did not significantly affect food intake or body weight gain (P > 0.05). However, the rats fed a

**Table 3. Food intake, final body weight, and average daily weight gain of rats fed the control or a high-fat diet with or without FCS.**

| Diet | FCS | | Mean ± S.E. | Two-way ANOVA P values | | |
|---|---|---|---|---|---|---|
| | (-) | (+) | | Diet | FCS | Diet × FCS |
| | **Total Food Intake (g/56 days)** | | | | | |
| CT | 1,020.51 ± 32.95 | 1,063.63 ± 33.00 | 1,042.07 ± 23.22[a] | 0.001 | 0.989 | 0.218 |
| HF | 893.98 ± 35.46 | 851.76 ± 32.83 | 872.87 ± 24.16[b] | | | |
| Mean | 957.25 ± 24.20 | 957.69 ± 23.26 | | | | |
| | **Food Intake (g/d)** | | | | | |
| CT | 18.22 ± 0.59 | 18.99 ± 0.59 | 18.61 ± 0.41[a] | 0.001 | 0.990 | 0.219 |
| HF | 15.96 ± 0.63 | 15.21 ± 0.59 | 15.59 ± 0.43[b] | | | |
| Mean | 17.09 ± 0.43 | 17.10 ± 0.42 | | | | |
| | **FBW (g)** | | | | | |
| CT | 386.20 ± 15.06 | 390.84 ± 15.14 | 388.52 ± 10.64[b] | 0.004 | 0.648 | 0.447 |
| HF | 459.93 ± 15.06 | 441.31 ± 15.03 | 450.62 ± 10.64[a] | | | |
| Mean | 423.07 ± 10.67 | 416.08 ± 10.67 | | | | |
| | **ADG (g/d)** | | | | | |
| CT | 4.80 ± 0.27 | 4.88 ± 0.27 | 4.84 ± 0.19[b] | 0.004 | 0.648 | 0.447 |
| HF | 6.12 ± 0.27 | 5.78 ± 0.27 | 5.95 ± 0.19[a] | | | |
| Mean | 5.46 ± 0.19 | 5.33 ± 0.19 | | | | |

Values are expressed as the mean ± S.E. (n = 7 in each group). Means in a column without a common letter differ significantly (P <0.05), according to the F test. CT (-): rats fed the control diet AIN-93G; CT (+): rats fed the control diet AIN-93G + FCS; HF (-): rats fed a high-fat diet; HF (+): rats fed a high-fat diet + FCS; FBW: final body weight; ADG: average daily weight gain.

high-fat diet (HF-) showed lower ($P < 0.05$) total and daily food intake and higher weight gain than those in rats fed the control diet (CT-).

## Hepatic and serum lipid concentration

The hepatic and serum lipid profiles were determined in rats fed the control or a high-fat diet, with or without the addition of FCS (Table 4). FCS intake caused a significant (P < 0.05) increase in serum HDL concentration, independent of the diet type. The high-fat diet promoted an increase in CLT and TG in the liver compared to the control diet.

**Table 4. Serum and hepatic lipid profile of rats fed the control or a high-fat diet with or without FCS.**

| Diet | FCS | | Mean ± S.E. | Two-way *ANOVA P* values | | |
|---|---|---|---|---|---|---|
| | (-) | (+) | | Diet | FCS | Diet×FCS |
| **CLT serum (mg/dL)** | | | | | | |
| CT | 70.22 ± 6.11 | 71.06 ± 7.53 | 70.64 ± 5.25 | 0.562 | 0.730 | 0.627 |
| HF | 67.67 ± 9.67 | 61.63 ± 7.83 | 64.65 ± 6.81 | | | |
| **Mean** | 68.94 ± 5.13 | 66.34 ± 4.96 | | | | |
| **CLT liver (mg/dL)** | | | | | | |
| CT | 8.90 ± 2.44 | 15.14 ± 2.52 | 12.02 ± 1.83[b] | 0.015 | 0.101 | 0.409 |
| HF | 18.57 ± 2.65 | 20.78 ± 2.37 | 19.68 ± 1.83[a] | | | |
| **Mean** | 13.74 ± 1.69 | 17.96 ± 1.69 | | | | |
| **TG serum (mg/dL)** | | | | | | |
| CT | 111.20 ± 16.30 | 96.71 ± 17.85 | 103.95 ± 12.57 | 0.869 | 0.362 | 0.965 |
| HF | 108.84 ± 17.21 | 92.90 ± 15.48 | 100.87 ± 11.78 | | | |
| **Mean** | 110.02 ± 10.97 | 94.80 ± 11.69 | | | | |
| **TG liver (mg/dL)** | | | | | | |
| CT | 79.34 ± 19.21 | 123.83 ± 19.82 | 101.59 ± 14.43[b] | 0.038 | 0.171 | 0.377 |
| HF | 146.69 ± 20.91 | 157.12 ± 18.69 | 151.91 ± 14.43[a] | | | |
| **Mean** | 113.03 ± 13.31 | 140.46 ± 13.31 | | | | |
| **HDL serum (mg/dL)** | | | | | | |
| CT | 63.62 ± 7.95 | 86.88 ± 8.15 | 75.26 ± 5.94 | 0.863 | 0.007 | 0.730 |
| HF | 59.26 ± 8.89 | 88.06 ± 7.67 | 73.66 ± 5.94 | | | |
| **Mean** | 61.44 ± 5.55[B] | 87.47 ± 5.55[A] | | | | |
| **HDL liver (mg/dL)** | | | | | | |
| CT | 7.53 ± 0.52 | 7.77 ± 0.54 | 7.65 ± 0.39 | 0.415 | 0.695 | 0.940 |
| HF | 8.06 ± 0.56 | 8.23 ± 0.50 | 8.14 ± 0.39 | | | |
| **Mean** | 7.80 ± 0.36 | 8.00 ± 0.36 | | | | |
| **LDL serum (mg/dL)** | | | | | | |
| CT | 21.70 ± 3.58 | 19.87 ± 3.44 | 20.79 ± 2.57 | 0.790 | 0.669 | 0.358 |
| HF | 17.33 ± 3.69 | 22.13 ± 3.40 | 19.73 ± 2.57 | | | |
| **Mean** | 19.51 ± 2.39 | 21.00 ± 2.39 | | | | |
| **LDL liver (mg/dL)** | | | | | | |
| CT | 4.55 ± 0.64 | 5.35 ± 0.61 | 4.95 ± 0.47 | 0.538 | 0.743 | 0.326 |
| HF | 4.69 ± 0.62 | 4.29 ± 0.64 | 4.49 ± 0.47 | | | |
| **Mean** | 4.62 ± 0.42 | 4.82 ± 0.42 | | | | |

Values are expressed as the mean ± S.E. (n = 5 in each group). CT (-): rats fed the control diet AIN-93G; CT (+): rats fed the control diet AIN-93G + FCS; HF (-) rats fed a high-fat diet; HF (+) rats fed a high-fat diet + FCS. Means in the same row without a common capital letter A, B differ (*P* < 0.05) and means in the same column without a common lowercase letter a, b differ (*P* < 0.05), according to the F test. CLT, total cholesterol; TG, triglyceride; HDL, high-density lipoprotein cholesterol; LDL, low-density lipoprotein cholesterol.

## Bile acid concentration in feces

The concentration of bile acids was determined in the feces of the rats fed the control and a high-fat diet, with or without the addition of FCS, at the first and eighth week of the treatment (Fig 1). Independent of the presence or absence of FCS in the diet, the concentration of bile acids in the rat feces was affected by the diet type and treatment duration (weeks). In the first and eighth week of the treatment, the fecal concentration of bile acids was significantly ($P < 0.05$) higher in the rats fed a high-fat diet than in the rats fed the control diet. The rats fed a high-fat diet also showed a significantly ($P < 0.05$) higher concentration of bile acids in their feces in the eighth week of the treatment than in the first week.

## Determination of fecal microbiota composition

The absolute quantification of specific groups of bacteria was determined in the feces of the rats fed the control or a high-fat diet, with or without the addition of FCS (Figs 2–5). The *P*-values of the statistical analyses are presented in S1 Table.

The fecal population of *Bacteroides* spp. only showed a diet × FCS × time interaction effect (Fig 2). In the first week of treatment, the population of *Bacteroides* spp. in feces of the rats fed a high-fat diet (HF-) was significantly ($P < 0.05$) lower than that in the other groups of rats, whereas the addition of FCS to high-fat diet (HF+) normalized the fecal *Bacteroides* spp. concentration (CT- × HF+; $P > 0.05$).

In the eighth week of the treatment, there was no significant effect of FCS addition on the population of *Bacteroides* spp. in feces of the rats fed the control or a high-fat diet.

The fecal population of *Bifidobacterium* spp. presented a diet × FCS interaction effect (Fig 3). The addition of FCS to high-fat diet (HF+) significantly increased ($P < 0.05$) the population of *Bifidobacterium* spp. in the rat feces compared to that in the rats fed a high-fat diet without FCS (HF-). The addition of FCS to the control diet did not affect the population of *Bifidobacterium* spp. (P > 0.05).

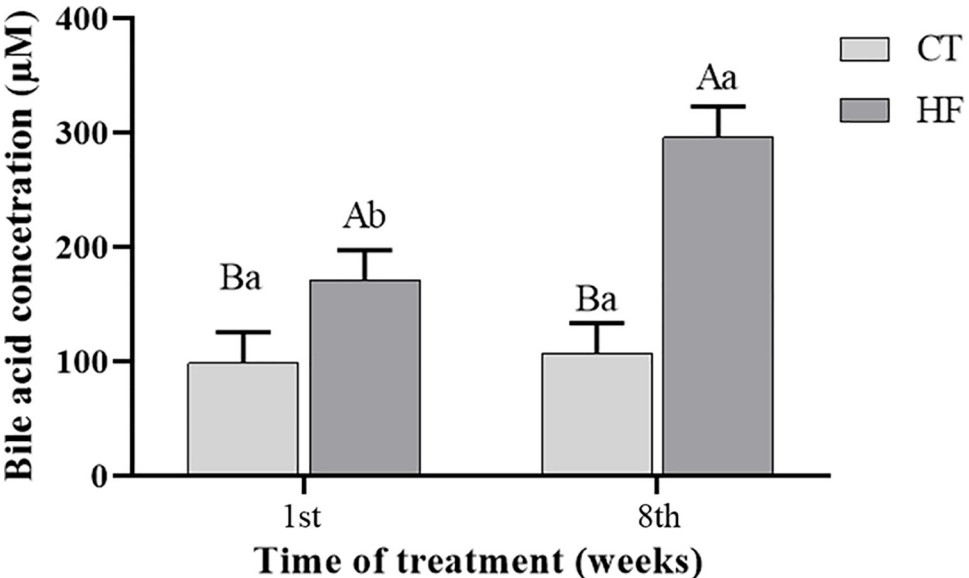

**Fig 1. Bile acid concentration in feces.** Values are least-square means of diet × treatment time interaction of bile acid concentration in feces ± S.E. (n = 5 in each group). CT: rats fed the control diet AIN-93G; HF: rats fed a high-fat diet. Means without a common capital letter A, B differ ($P < 0.05$) in relation to the effect of diet type on treatment duration and means without a common lowercase letter a, b differ ($P < 0.05$) in relation to the effect of treatment duration on diet type, according to the F test.

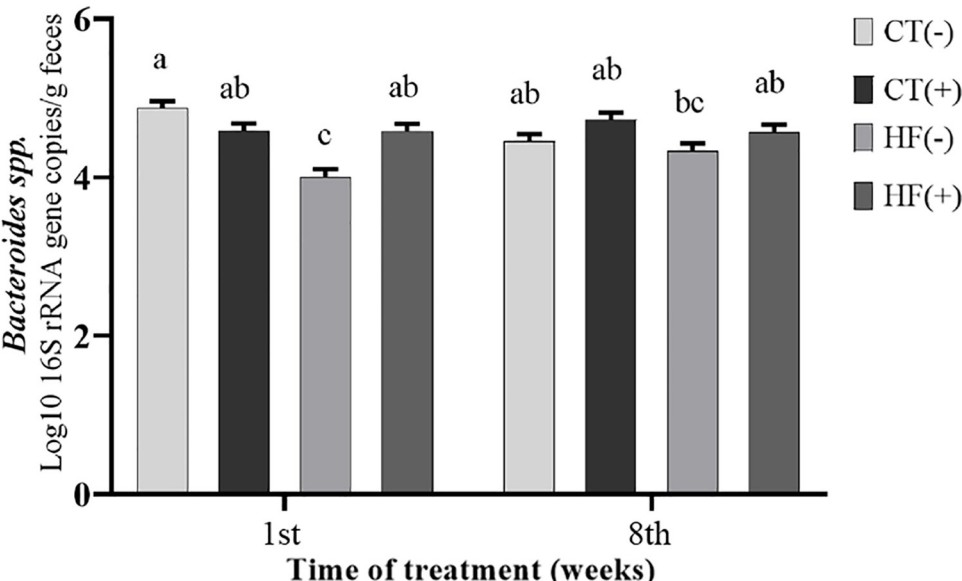

**Fig 2. *Bacteroides* spp. population in fecal samples.** Values are least-square means of the diet × FCS × treatment duration interaction of *Bacteroides* spp. population in the fecal samples of rats fed the control diet without FCS [CT (-)], control diet + FCS [CT (+)], high-fat diet [HF (-)], or high-fat diet + FCS [HF (+)]. Data represent the mean log10 16S rRNA gene copy number per gram of feces. Values are expressed as the mean ± S.E. (n = 5 in each group). Means without a common lowercase letter differ ($P < 0.05$), according to the Tukey test.

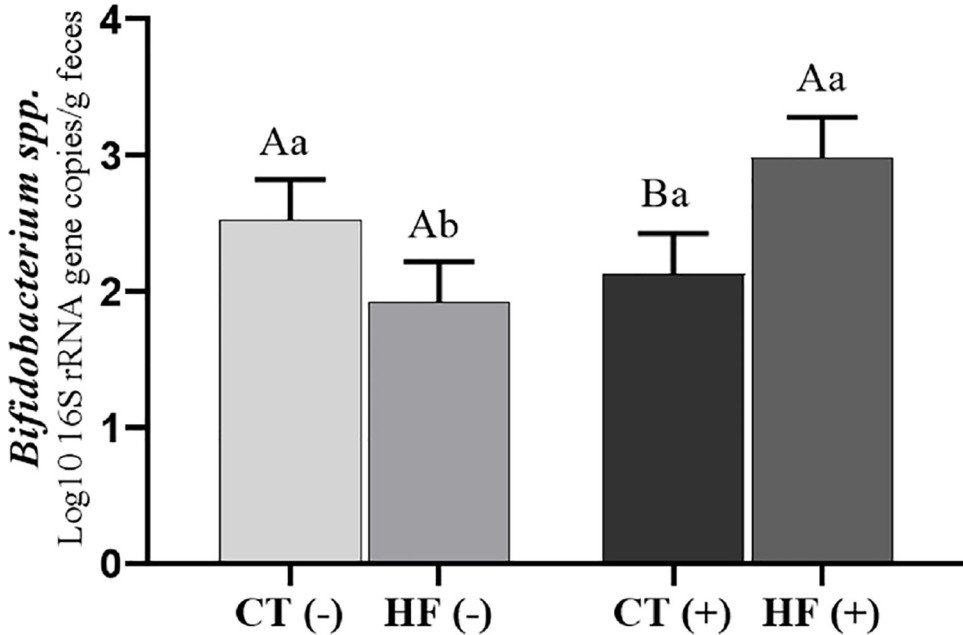

**Fig 3. *Bifidobacterium* spp. population in fecal samples.** Values are least-square means of the diet × FCS interaction of *Bifidobacterium* spp. population in the fecal samples of rats fed the control diet without FCS [CT (-)], high-fat diet [HF (-)], control diet + FCS [CT (+)], or high-fat diet + FCS [HF (+)]. Data represent the mean log10 16S rRNA gene copy number per gram of feces. Values are expressed as the mean ± S.E. (n = 5 in each group). Means without a common capital letter A, B differ ($P < 0.05$) in relation to the effect of diet type on coffee consumption and means without a common lowercase letter a, b differ ($P < 0.05$) in relation to the effect of coffee consumption on diet type, according to the F test.

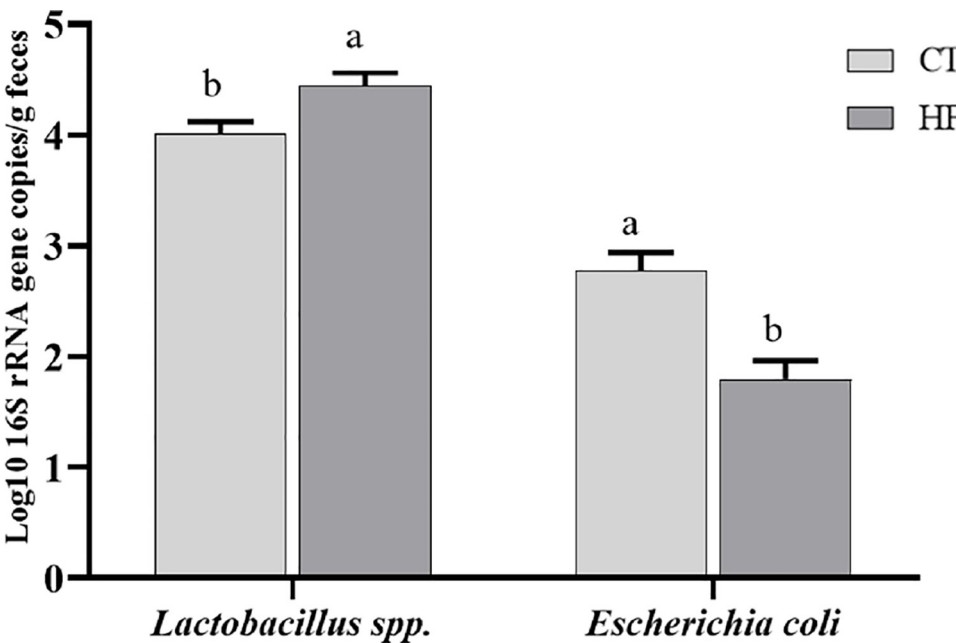

**Fig 4. Populations of *Lactobacillus* spp. and *E. coli* in the fecal samples of rats fed the control or a high-fat diet.**
Data represent the mean log10 16S rRNA gene copy number per gram of feces. Values are expressed as the mean ± S.E.
(n = 5 in each group). Within the same bacterial strain, means in the same column without a common letter differ
significantly ($P < 0.05$), according to the F test.

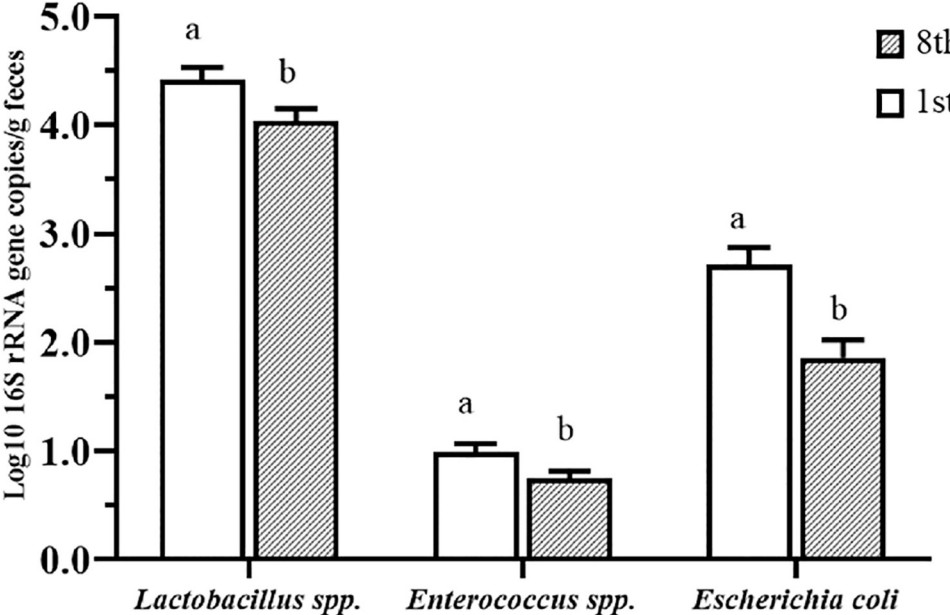

**Fig 5. Populations of *Lactobacillus* spp., *E. coli*, and *Enterococcus* spp. in the fecal samples.** Rats were fed the
control or a high-fat diet, with or without FCS, after first and eighth week of the treatment. Data represent the mean
log10 16S rRNA gene copy number per gram of feces. Values are expressed as the mean ± S.E. (n = 5 in each group).
Within the same bacterial strain, means in the same column without a common letter differ significantly ($P < 0.05$),
according to the F test.

The high-fat diet significantly affected ($P < 0.05$) the populations of *Lactobacillus* spp. and *E. coli* in the feces of rats, regardless of the presence of FCS or the treatment duration (Fig 4). The *Lactobacillus* spp. population was significantly ($P < 0.05$) higher and *E. coli* population was significantly ($P < 0.05$) lower in the feces of rats fed a high-fat diet (HF) than those in the feces of rats fed the control diet.

The treatment duration (weeks) with the experimental diet significantly affected ($P < 0.05$) the populations of *Lactobacillus* spp., *Enterococcus* spp., and *E. coli*, regardless of the dietary fat content or the addition of FCS to the diet (Fig 5). The populations of *Lactobacillus* spp., *Enterococcus* spp., and *E. coli* were significantly ($P < 0.05$) higher in the feces of rats in the first week of treatment with the experimental diet than those in the eighth week.

## Microbiota composition of the cecum content

The populations of *Enterococcus* spp., *Bifidobacterium* spp., and *E. coli* present in the rat cecum content were significantly affected ($P < 0.05$) by the addition of FCS to the diet (Table 5). Compared to the control diet alone (CT-), the *Enterococcus* spp. population in the cecum content was significantly increased ($P < 0.05$) with the addition of FCS to the control diet (CT+). Compared to the high-fat diet alone (HF-), the population of *Bifidobacterium* spp. significantly increased ($P < 0.05$) when FCS was added to the high-fat diet (HF+); however, no difference was observed in the *Bifidobacterium* spp. population when FCS was added to the

**Table 5. Bacterial populations in the cecum content of rats fed the control or a high-fat diet with or without FCS.**

| Diet | FCS | | Mean ± S.E. | Two-way *ANOVA P* values | | |
|---|---|---|---|---|---|---|
| | (-) | (+) | | Diet | FCS | Diet×FCS |
| | log10 16S rRNA gene copy number per gram of feces | | | | | |
| | *Bacteroides* spp. | | | | | |
| CT | 4.68 ± 0.10 | 4.78 ± 0.10 | 4.73 ± 0.07 | 0.610 | 0.486 | 0.784 |
| HF | 4.64 ± 0.11 | 4.69 ± 0.10 | 4.67 ± 0.08 | | | |
| Mean | 4.66 ± 0.07 | 4.73 ±0.07 | | | | |
| | *Lactobacillus* spp. | | | | | |
| CT | 3.51 ± 0.14 | 3.85 ± 0.12 | 3.68 ± 0.10[b] | 0.001 | 0.338 | 0.167 |
| HF | 4.41 ± 0.16 | 4.33 ± 0.12 | 4.37 ± 0.10[a] | | | |
| Mean | 3.96 ± 0.09 | 4.09 ± 0.09 | | | | |
| | *Enterococcus* spp. | | | | | |
| CT | 0.27 ± 0.14[Bb] | 0.70 ± 0.11[Aa] | 0.49 ± 0.10 | 0.037 | 0.194 | 0.036 |
| HF | 0.93 ± 0.13[Aa] | 0.81 ± 0.11[Aa] | 0.87 ± 0.09 | | | |
| Mean | 0.61 ± 0.08 | 0.75 ± 0.07 | | | | |
| | *Bifidobacterium* spp. | | | | | |
| CT | 1.77 ± 0.25[Aa] | 2.12 ± 0.26[Aa] | 1.94 ± 0.19 | 0.370 | 0.001 | 0.015 |
| HF | 1.33 ± 0.28[Ba] | 3.09 ± 0.24[Ab] | 2.21 ± 0.19 | | | |
| Mean | 1.55 ± 0.17 | 2.61 ± 0.17 | | | | |
| | *E. coli* | | | | | |
| CT | 1.82 ± 0.20[Ba] | 3.02 ± 0.21[Aa] | 2.42 ± 0.15 | 0.003 | 0.263 | 0.003 |
| HF | 1.73 ± 0.22[Aa] | 0.98 ± 0.19[Bb] | 1.36 ± 0.15 | | | |
| Mean | 1.78 ± 0.13 | 2.00 ± 0.13 | | | | |

Values are expressed as the mean ± S.E. (n = 5 in each group). CT (-): rats fed the control diet AIN-93G; CT (+): rats fed the control diet AIN-93G + FCS; HF (-): rats fed a high-fat diet; HF (+): rats fed a high-fat diet + FCS. Means in the same row without a common capital letter A, B differ significantly ($P < 0.05$), and means in the same column without a common lowercase letter a, b differ significantly ($P < 0.05$), according to the F test.

control diet. The *E. coli* population in the cecum content showed a significant increase ($P < 0.05$) when FCS was added to the control diet, while a significant reduction ($P < 0.05$) was observed when FCS was mixed with the high-fat diet, compared to the CT- and HF- diets, respectively. The *Enterococcus* spp. population was not significantly affected ($P > 0.05$) by the diet type in which FCS was incorporated. Compared to the control diet (CT-), the high-fat diet (HF-) promoted an increase in the *Enterococcus* spp. population in the cecum content and did not affect *Bifidobacterium* spp. and *E. coli* populations.

The populations of *Bacteroides* spp. and *Lactobacillus* spp. in the cecum content were not significantly affected ($P > 0.05$) by the addition of FCS, independent of the diet type. However, the high-fat diet promoted a significant increase ($P < 0.05$) in the population of *Lactobacillus* spp. in the rat cecum content.

## Expression of pro-inflammatory genes in the large intestine

The mRNA expression levels of Il1b, Il6, and Tnfa were evaluated in rats fed the control and a high-fat diet, with and without FCS (Table 6). Compared to the high-fat diet alone (HF-), the addition of FCS to the high-fat diet (HF+) downregulated ($P < 0.05$) the mRNA expression of Il1b in the large intestine, whereas no difference was observed between the CT- and CT + groups. Regarding mRNA expression level of Il6 in the large intestine, addition of FCS increased this level regardless of the diet type, whereas the high-fat diet promoted a decrease in the level, independent of whether or not FCS was used. In contrast, Tnfa mRNA level in the large intestine was not significantly affected ($P > 0.05$) by the diet type or the presence of FCS.

## Discussion

This study evaluated the effect of FCS when mixed with the diet on the physiological parameters, lipid profile, bile acid concentration, and microbiota of rats in a high-fat diet-induced obesity model.

Several studies have shown the effectiveness of coffee extract in reducing body weight gain in rats fed a high-fat diet [9, 38–42]. Polyphenols, which are present in high concentration in

**Table 6. Quantification of Il1b, Il6, and Tnfa mRNA levels in the large intestine of rats fed the control or a high-fat diet with or without FCS.**

| Diet | FCS | | Mean ± S.E. | Two-way *ANOVA P* values | | |
|---|---|---|---|---|---|---|
| | (-) | (+) | | Diet | FCS | Diet×FCS |
| | mRNA Il1b | | | | | |
| CT | $1.00 \pm 0.09^{Aa}$ | $1.19 \pm 0.09^{Aa}$ | $1.10 \pm 0.06$ | 0.068 | 0.049 | 0.006 |
| HF | $1.21 \pm 0.10^{Aa}$ | $0.63 \pm 0.10^{Bb}$ | $0.91 \pm 0.07$ | | | |
| Mean | $1.11 \pm 0.06$ | $0.91 \pm 0.06$ | | | | |
| | mRNA Il6 | | | | | |
| CT | $1.02 \pm 0.23$ | $1.62 \pm 0.21$ | $1.32 \pm 0.16^{a}$ | 0.025 | 0.018 | 0.825 |
| HF | $0.53 \pm 0.19$ | $1.04 \pm 0.21$ | $0.90 \pm 0.14^{b}$ | | | |
| Mean | $0.78 \pm 0.15^{B}$ | $1.33 \pm 0.15^{A}$ | | | | |
| | mRNA Tnfa | | | | | |
| CT | $0.97 \pm 0.22$ | $0.87 \pm 0.22$ | $0.92 \pm 0.15$ | 0.849 | 0.580 | 0.287 |
| HF | $0.72 \pm 0.18$ | $1.05 \pm 0.18$ | $0.88 \pm 0.13$ | | | |
| Mean | $0.85 \pm 0.14$ | $0.96 \pm 0.14$ | | | | |

Values are expressed as the mean ± S.E. (n = 6 in each group). CT (-): rats fed the control diet AIN-93G; CT (+): rats fed the control diet AIN-93G + FCS; HF (-): rats fed a high-fat diet; HF (+): rats fed a high-fat diet + FCS. Means in the same row without a common capital letter A, B differ significantly ($P < 0.05$), and means in the same column without a common lowercase letter a, b differ significantly ($P < 0.05$), according to the F test.

coffee, can inhibit digestive enzymes [43], and consequently, inhibit macronutrient absorption and reduce body weight gain [44, 45]. Contrary to these observations, in the present study, the consumption of FCS for 56 days did not change body weight of rats in the control group or high-fat diet group. This result suggests that in a treatment model for preventing obesity, FCS should be co-administered during the development of obesity for a longer duration to observe its effect on body weight. A previous study that monitored obesity induction over 83 days by feeding rats a high-fat diet showed that although rats on the high-fat diet had a higher body weight than that of the control rats after 21 days of treatment, statistically significant difference was observed only after 83 days [46]. This reinforces the hypothesis that the rats were in a pre-obese state in the present study. Furthermore, several studies that have observed significant effects of coffee on body weight [9, 39, 41] used an obesity treatment model instead of an obesity prevention model of the present study.

The higher body weight of the rats fed the high-fat diet despite their lower food intake was associated with the higher energy density of the high-fat diet (5.30 kcal/g) than that of the control diet (3.95 kcal/g). Although the dietary intake of rats fed the high-fat diet was 16% less than that of the control rats, their energy intake was 13% greater. Similar results have been reported in other studies [47–49]. The authors suggest that the decrease in food intake promoted by the high-fat diet may be attributed to a compensatory mechanism that maintains energy balance homeostasis and consequently controls the excess body weight gain induced by high caloric density of the diet.

Several other factors such as coffee bean species (*C. arabica* and *C. canephora* var. *Robusta*), roasting process (time × temperature) [50], and method used to prepare the beverage influence coffee polyphenol content. Rendón *et al.* [51] showed that filtered coffee beverages have lower diterpene content than unfiltered beverages. According to Cruz *et al.* [52], variations in the biological activity of coffee may be related to differences in its chemical composition.

In the present study, the coffee solution was freeze-dried to enable its incorporation into a high-fat diet. Although this process may promote alterations in the chemical composition, such modifications can be considered minimal, as freeze-drying is the most recommended method for preserving polyphenol compounds [53, 54]. Moreover, the dose of FCS used in the present study was 0.39 g/100 g of diet, which is equivalent to 163 mL/day or 1.5 cups/day of coffee, the average coffee intake of the Brazilian population [20]. When testing two doses of coffee extract (40% and 60% *v/v*), Maki *et al.* [55] observed a significant reduction in body weight gain and amelioration of some biochemical markers in the mice treated with the high dose (60%) of coffee extract. Therefore, it is possible that the dose of FCS used in the present study was insufficient to reduce the body weight of rats.

Regarding the lipid profile, there is no consensus concerning the effect of coffee [41, 56–58]. In the present study, the consumption of FCS did not affect most lipid profile markers in the serum and liver, since the levels of cholesterol, TG, and LDL were similar between the control and high-fat diet groups, with and without the addition of FCS. Similar to our results, Karabudak *et al.* [59] demonstrated no significant association between Turkish or instant coffee consumption and serum lipid profile in subjects. The authors suggested that the preparation method and the amount of coffee consumed are important aspects that influence the serum lipid response. The cause of these differences is not clear but may be associated with the amount of coffee administered, different routes of coffee administration (gavage or mixed with the diet), and different treatment protocols, as some studies have employed an obesity treatment protocol rather than an obesity prevention protocol. Ilmiawati *et al.* [42] observed a decrease in serum CLT and TG levels with low doses of a green coffee extract; however, only high doses of green coffee extract (> 20 mg/kg body weight/day) were able to decrease LDL level and increase HDL level in the serum.

Although FCS did not decrease the levels of CLT or LDL, an atherogenic lipoprotein, coffee consumption promoted a significant increase in the serum HDL level. This result shows that coffee has atheroprotective property, as HDL lipoproteins remove excess cholesterol from the tissues by reverse cholesterol transport [60]. This is also in line with the marginal increase in CLT in the liver observed with coffee treatment (1.31-fold compared to no coffee; $P = 0.100$). Similar to our results, Feyisa *et al*. [56] observed that rats fed coffee had higher HDL-C concentration than that in other rats. Some studies, most of which used purified compounds, showed that some polyphenols found in coffee (caffeic, ferulic, and phenolic acids) as well as in other foods stimulated reverse cholesterol transport by promoting HDL formation and cholesterol efflux [61, 62].

Contrarily, no change was observed in the serum lipid profile and only the hepatic concentrations of cholesterol and TG increased in rats fed the high-fat diet. We hypothesized that the obesity induction time was not sufficient to surpass systemic homeostasis; therefore, excess cholesterol was taken up by the liver through LDL receptors mediated by the SCAP-SREBP pathway [63], which caused an increase in hepatic cholesterol, while serum levels remained the same compared to the control rats. The higher bile acid excretion in stools observed in the first week of treatment, which was even greater in the eighth week, in rats fed the high-fat diet regardless of FCS consumption, supports the hypothesis that systemic homeostasis was maintained until 56 days of treatment.

Bile acids are amphipathic sterols secreted in the duodenum. They are the main constituents of bile and are responsible for emulsifying fat and facilitating digestion and absorption [64]. As expected, the high-fat diet promoted an increase in the total bile acid content in the rat feces. Lin *et al*. [65] observed lower concentrations of conjugated bile acids and higher levels of unconjugated bile acids in the feces of rats fed a high-fat diet than those in other rats. Gut microbiota can hydrolyze conjugated bile acids into secondary bile acids, which are associated with colon cancer [66, 67]. Although coffee and its polyphenolic compounds can decrease dietary lipid digestion by reducing the synthesis and action of bile acids [60, 68], in the present study, FCS did not affect total bile acid excretion in the feces.

Certain strains of the genus *Bifidobacterium* are involved in anti-obesity activity and are therefore found in low number in obese individuals. The FCS-induced increase in *Bifidobacterium* spp. population in the feces and cecal content only in rats fed the high-fat diet suggests that the consumption of FCS during obesity development and progression may have a protective effect; additionally, these rats presented lower mRNA level of Il1b in the large intestine than those fed the high-fat diet. These effects may be associated with high polyphenol content of coffee (chlorogenic acids, caffeine, cafestol, and kahweol), which have anti-inflammatory properties [69, 70]. Some studies have shown that polyphenols in foods have prebiotic effect, favoring the *Bifidobacterium* spp. population in the gut [71, 72]. Fermentation products of coffee components may inhibit the growth of other microorganisms in the colon, and consequently, favor an increase in the *Bifidobacterium* spp. population in rats fed coffee combined with a high-fat diet. Nakayama and Oishi [30] showed an increase in the *Bifidobacterium* spp. population in the feces of rats fed coffee extract, thus corroborating our results.

Although the genus *Lactobacillus* has many health benefits, some subspecies appeared to be positively associated with body weight gain [73]. The high body weight gain observed in rats fed the high-fat diet in the present study may be associated with the presence of some subspecies of the genus *Lactobacillus*, generally observed in obese organisms, since a significant increase in the *Lactobacillus* spp. population was observed in the feces of these rats. Similar to our results, Cowan *et al*. [9] did not observe any effect of coffee intake on the fecal population of *Lactobacillus* spp.

*E. coli* is generally found in the human gastrointestinal tract, either as commensal, probiotic, or pathogenic bacteria [74], comprising less than 0.1% of the total bacterial count in the gut microbiota [75]. The present study observed a reduction in the fecal *E. coli* population in rats fed the high-fat diet, independent of the presence or absence of FCS in the diet. Corroborating our findings, Million *et al.* [76] suggested that the absence of *E. coli* was an independent predictor of weight gain, as they found *E. coli* in the feces of patients with weight loss. *In vitro* studies have shown that phenolic compound extracts modulated *E. coli* growth [77–79]. Therefore, in the present study, coffee polyphenols or the products of their hydrolysis or reduction may have stimulated the growth of *E. coli* in the cecum of rats fed the control diet. In line with data from the literature, *Lactobacillus* spp., *E. coli*, and *Enterococcus* spp. populations decreased in the feces with increase in the treatment duration, independent of the diet type. Considering the high density and immense diversity of microorganisms in the gastrointestinal tract, particularly in the colon, the release and/or production of different secondary metabolites by these microorganisms (organic acids and short-chain fatty acids) may explain the reduction in some bacterial strains, since these compounds act as antimicrobial agents [80, 81]. Therefore, the longer treatment duration may have increased the antimicrobial action of some metabolites against *Lactobacillus*, *E. coli*, and *Enterococcus*, reducing these bacterial populations in the feces.

*Lachnospira*, *Roseburia*, *Butyrivibrio*, *Ruminoccus*, *Fecalibacterium*, and *Fusobacteria* constitute the cecal microbiota [82]. However, in the present study, *E. coli*, *Enterococcus* spp., *Lactobacillus* spp., *Bacteroides* spp., and *Bifidobacterium* spp. were identified in the cecum. According to Liu *et al.* [44], *Enterococcus* spp. and *Bifidobacterium* spp. hydrolyze food polyphenols. Therefore, the observed increase in the populations of *Bifidobacterium* spp. and *Enterococcus* spp. in the cecum of rats fed FCS mixed with the control diet and FCS mixed with the high-fat diet compared to rats fed the control and high-fat diets alone, respectively, may be associated with polyphenols present in FCS. In addition to the effect of FCS, it is likely that some fecal strains of *Bifidobacterium* spp. are resistant to bile acids, which results in an increase in its population. In some strains such as *B. longum* and *B. brevis*, resistance mechanisms to bile acids involving bile efflux systems have already been identified [83–85].

Considering the increase in the levels of pro-inflammatory cytokines such as IL-6, IL-1β, and TNF-α in obese individuals [86] and the decrease in the serum level of IL-6 after consumption of four or more cups of coffee/day [87], we determined the mRNA level of Il6 in the large intestine of rats to investigate a possible coffee × fecal microbiota × gut inflammatory response interaction. Despite the high polyphenol content in coffee, Il6 mRNA level was increased after coffee consumption regardless of the diet type, whereas the high-fat diet decreased Il6 mRNA level in the large intestine. The high-fat diet-induced decrease in Il6 mRNA level in the large intestine may be related to the inhibition of the *E. coli* population observed in these rats compared to the control. According to Kittana *et al.* [88], a greater population of commensal *E. coli* was observed in the intestinal tissues of humans during inflammation. In addition, they demonstrated that some strains of *E. coli* in the intestine are associated with high secretion of IL-6.

Despite being reported that foods rich in polyphenols inhibit pro-inflammatory cytokine secretion [89], in the present study, coffee promoted an increase in Il6 mRNA level in the large intestine regardless of the diet type. This contradiction may be attributed to differences in the tissue response. Most studies evaluated systemic inflammation by measuring serum levels of these cytokines, while in the present study, we evaluated the inflammatory response in the large intestine. According to Juge-Aubry *et al.* [90], IL-6 may be involved in anti-inflammatory activity, since it can decrease TNF-α and interferon gamma (IFNγ) levels during inflammation, and therefore, control inflammation and inhibit tissue damage. Caro-Gómez *et al.* [91]

observed increased expression level of hepatic IL-6 in rats fed green coffee extract. They suggested that IL-6 can control obesity-associated inflammation by favoring macrophage polarization towards the M2 phenotype, which acts in the resolution phase of inflammation and in repairing damaged tissues.

## Conclusions

The results suggest that the consumption of FCS may promote positive health effects in rats fed a high-fat diet by increasing the populations of *Bifidobacterium*, improving HDL-C reverse cholesterol transport, and reducing Il1b mRNA level. However, FCS administration at a dose equivalent to the regular average coffee intake of the Brazilian population over an eight-week period was not effective in controlling food intake, and consequently, preventing weight gain in rats of high-fat diet-induced obesity model. Therefore, coffee, being associated with a wide range of health benefits, must be consumed as filtered beverage regularly in moderate amount (3–5 cups/day).

## Supporting information

**S1 Table. *P*-values from the statistical analysis of data of microbial composition and bile acid concentration in the feces of rats fed the control diet without FCS [CT (-)], control diet + FCS [CT (+)], high-fat diet [HF (-)], or high-fat diet + FCS [(HF +)].** D: diet; FCS: freeze-dried coffee solution; T: time; SEM: standard error of the mean.
(DOCX)

**S1 Graphical abstract.**
(TIF)

## Author Contributions

**Conceptualization:** Marilia Hermes Cavalcanti, Eliana dos Santos Leandro, Sandra Fernandes Arruda.

**Data curation:** Marilia Hermes Cavalcanti, João Paulo Santos Roseira, Eliana dos Santos Leandro, Sandra Fernandes Arruda.

**Formal analysis:** Marilia Hermes Cavalcanti, João Paulo Santos Roseira, Eliana dos Santos Leandro, Sandra Fernandes Arruda.

**Funding acquisition:** Eliana dos Santos Leandro, Sandra Fernandes Arruda.

**Investigation:** Marilia Hermes Cavalcanti, João Paulo Santos Roseira, Eliana dos Santos Leandro, Sandra Fernandes Arruda.

**Methodology:** Marilia Hermes Cavalcanti, João Paulo Santos Roseira, Eliana dos Santos Leandro, Sandra Fernandes Arruda.

**Project administration:** Eliana dos Santos Leandro, Sandra Fernandes Arruda.

**Resources:** Eliana dos Santos Leandro, Sandra Fernandes Arruda.

**Software:** Marilia Hermes Cavalcanti, João Paulo Santos Roseira.

**Supervision:** Eliana dos Santos Leandro, Sandra Fernandes Arruda.

**Validation:** Marilia Hermes Cavalcanti, João Paulo Santos Roseira, Sandra Fernandes Arruda.

**Visualization:** Marilia Hermes Cavalcanti, João Paulo Santos Roseira, Eliana dos Santos Leandro, Sandra Fernandes Arruda.

**Writing – original draft:** Marilia Hermes Cavalcanti, João Paulo Santos Roseira, Eliana dos Santos Leandro, Sandra Fernandes Arruda.

**Writing – review & editing:** Marilia Hermes Cavalcanti, João Paulo Santos Roseira, Eliana dos Santos Leandro, Sandra Fernandes Arruda.

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
