## [Decision Letter · Decision Letter 0]

14 Sep 2021

PONE-D-21-14462Effect of freeze-dried coffee solution in a high-fat diet-induced obesity model in rats: biochemical and inflammatory impacts and effects on gut microbiotaPLOS ONE

Dear Dr. Cavalcanti,

Thank you for submitting your manuscript to PLOS ONE. After careful consideration, we feel that it has merit but does not fully meet PLOS ONE’s publication criteria as it currently stands. Therefore, we invite you to submit a revised version of the manuscript that addresses the points raised during the review process.

In addition to the reviewer's comments found below. Please also address the following editorial comments:Abstract should be revised and detailed methods removed.Rationale for targeted bacterial taxa should be provided in the introduction.Table 1, 2 – why are the values for the 2 treatments averaged? This data should be removed. Please provide ANOVA data (F and df) for main effects. What posthoc comparison was used?Please submit your revised manuscript by Oct 29 2021 11:59PM. If you will need more time than this to complete your revisions, please reply to this message or contact the journal office at plosone@plos.org. Please include the following items when submitting your revised manuscript:A rebuttal letter that responds to each point raised by the academic editor and reviewer(s). You should upload this letter as a separate file labeled 'Response to Reviewers'.A marked-up copy of your manuscript that highlights changes made to the original version. You should upload this as a separate file labeled 'Revised Manuscript with Track Changes'.An unmarked version of your revised paper without tracked changes. You should upload this as a separate file labeled 'Manuscript'.

We look forward to receiving your revised manuscript.

Kind regards,

Jane Foster, PhD

Academic Editor

PLOS ONE

Journal Requirements:

Reviewers' comments:

Reviewer's Responses to Questions

**Comments to the Author**

1. Is the manuscript technically sound, and do the data support the conclusions?

Reviewer #1: Partly

Reviewer #2: No

Reviewer #3: Partly

2. Has the statistical analysis been performed appropriately and rigorously? 

Reviewer #1: Yes

Reviewer #2: Yes

Reviewer #3: No

3. Have the authors made all data underlying the findings in their manuscript fully available?

Reviewer #1: No

Reviewer #2: Yes

Reviewer #3: No

4. Is the manuscript presented in an intelligible fashion and written in standard English?

Reviewer #1: Yes

Reviewer #2: No

Reviewer #3: Yes

5. Review Comments to the Author

Reviewer #1: 1. INTRODUCTION 1: Paragraph of introduction should reveal the strength and interest topic as a background why this topic coffee: constituents and health benefit discussed. The tradition of coffee consumption can be the reason for this discussion (include: what, who, where, when, why and how). Introduction should decide about specific topic that will be discussed on the paper. Started from general facts and issues as a background, to the specific issues.

2. INTRODUCTION 2: Paragraph should be arranged in a continuous and related way, for example in one paragraph what is an explanatory sentence is really explaining the main sentence. The main sentence explains about obesity in rat, but experiment only made the rats severe dyslipidemia. Explaining about do they severe obesity after given high fat diet. the research should show that the coffee method serving affected to the body weight. You need to concern with body weight (as your reason for obesity) or only dylipedemia by testing profil lipid level on your research.

3. METHOD: animal weight before experiment were in similar condition (normal)

4. Conclusion: Add more about the negativity of excessive coffee consumption in the last line so readers know about the importance of limiting coffee consumption in addition to the benefits that might arise. Add more about contraindication of coffee consumption (positivity and negativity) for each methods as a suggestion for readers which method of the coffee they will consume.

Reviewer #2: The abstract is too long, please short it.

Most of the data show that FCS was not effective in controlling obesity. I doubt that the consumption of FCS may have no positive health effect in rats fed with a high-fat diet.

Reviewer #3: In this study, freeze-dried coffee solution consumption was used to investigate physiological parameters, lipid profile, and microbiota during obesity induction in rats by a high-fat diet. The language of the manuscript is not very good and needs to be strengthened. References must be refreshed, some of them are too old. Results seem to be free from apparent manipulation, it is necessary to compare yours with similar studies. The sequence number of the figure is confused. How to conduct dietary intervention for animals? How to determine that the provided feed has been completely consumed by the animal? How to determine the amount of freeze-dried coffee added? Why choose 3.9 g?

6. PLOS authors have the option to publish the peer review history of their article (what does this mean?). If published, this will include your full peer review and any attached files.

Reviewer #1: **Yes: **Arie Dwi Alristina

Reviewer #2: No

Reviewer #3: No

---

## [Author Response · Author response to Decision Letter 0]

21 Oct 2021

Brasilia, October 18, 2021

To: Jane Foster, PhD

 Academic Editor

 PLOS ONE

Subject: Revised version of the manuscript PONE-D-21-14462

Dear Ph.D. Jane Foster,

Thank you for the opportunity to submit a revised version of our manuscript “Effect of a freeze-dried coffee solution in a high-fat diet-induced obesity model in rats: biochemical and inflammatory impacts and effects on gut microbiota”. Below are the responses to each point brought up by reviewers #1, #2, and #3. Some changes made were marked in the revised manuscript copy and the answers to some reviewer’s questions are presented below. We appreciate and thank the reviewers for their comments and suggestions.

Yours sincerely,

Marilia Hermes Cavalcanti

Postgraduate Program in Human Nutrition, Faculty of Health Sciences, Campus Universitário Darcy Ribeiro, Universidade de Brasília, Brasília, POBox 70910-900, Brazil. Phone +(55) - 61 - 31070092 e-mail: mariliaunb@outlook.com

In addition to the reviewer's comments found below. Please also address the following editorial comments:

1. Abstract should be revised and detailed methods removed.

The abstract was revised. 

2. Rationale for targeted bacterial taxa should be provided in the introduction.

A summary paragraph describing the rationale for the targets bacterial taxa were included in the introduction section. Page 4, line 79-92 (file ‘Manuscript’).

3. Table 1, 2 – why are the values for the 2 treatments averaged? This data should be removed. Please provide ANOVA data (F and df) for main effects. What posthoc comparison was used?

We consider that these comments refer to tables 3 and 4, not to tables 1 and 2 which present the sequence of primers. The values presented in tables 3 and 4 refer to the least-square means estimated by the SAS for the combination of factors and each studied factor (coffee and diet). After performing the analysis of variance using the F test, when analyzing the p-value of the variables expressed in tables 3 and 4, it was found that there was no significant effect for the interaction of the studied factors (coffee ᵡ diet). There was only a significant effect (p < 0.05) for one of the factors or there was no significant effect (p >0.05) for the factors studied on the variables presented in tables 3 and 4. Therefore, the letters that indicate the differences of only one factor must be overwritten in the estimated least-squares means for each factor, regardless of the interaction. As within each factor, there are only two levels (coffee, with (+) or without (-) and diet, control (CT+) or high-fat diet (HF+), the F test itself demonstrates the difference, there is no need for a posthoc test, as shown in tables 3 and 4.

Reviewers' comments:

Reviewer #1: 

1. INTRODUCTION 1: Paragraph of introduction should reveal the strength and interest topic as a background why this topic coffee: constituents and health benefit discussed. The tradition of coffee consumption can be the reason for this discussion (include: what, who, where, when, why, and how). Introduction should decide about specific topic that will be discussed on the paper. Started from general facts and issues as a background, to the specific issues.

The introduction has been extensively modified and organized according to the reviewer suggestions. 

2. INTRODUCTION 2: Paragraph should be arranged in a continuous and related way, for example in one paragraph what is an explanatory sentence is really explaining the main sentence. The main sentence explains about obesity in rat, but experiment only made the rats severe dyslipidemia. Explaining about do they severe obesity after given high fat diet. the research should show that the coffee method serving affected to the body weight. You need to concern with body weight (as your reason for obesity) or only dyslipidemia by testing profile lipid level on your research.

The introduction has been extensively modified and organized according to the reviewer suggestions. 

3. METHOD: animal weight before experiment were in similar condition (normal)

The animals used in the experiment had an initial mean body weight of 67.37 ± 6.04g. The precision of an experiment can be increased considerably by equalizing potential sources of error between the different treatments to be compared. Thus, the alternative is to measure the factors that are relevant to the precision of the experiment in order to try to correct the influence exerted on the response variable (Fisher, 1934).

In this context, analysis of covariance allows adjusting the effect of a response variable that was influenced by a variable or an uncontrolled source of variation, combining two widely applied procedures: analysis of variance (ANOVA) and regression (Fisher, 1934). The variable measured in the initial condition of the experimental unit is an auxiliary variable, also called a concomitant variable or covariate (Cochran, 1957). It is important to emphasize that the covariate needs to be correlated with the response variable and it must be ensured that it is not affected by the treatment so that this analysis can be used. As an example, there is the situation in which there are animals with different initial weights and a response variable of interest is the final weight of the animals (Fisher, 1934).

Thus, the consideration of some predictable effects, such as the initial weight of the animal, is intended to allow the animals to be compared and evaluated under equal conditions, since the final weight of the animals, in this study, is a variable response of interest. 

The use of covariates in studies allows to increase the precision of randomized experiments, clarify the nature of treatment effects, adjust regressions in multiple classifications, among other advantages (Cochran, 1957).

 In the materials and methods section, page 11 line 255-256 (file ‘Manuscript’) is described that the animals' initial weight was used as a covariate in the statistical analysis.

Cochran, W. G., 1957. Analysis of covariance: Its nature and uses. Biometrics 13 (3): 261–281.

Fisher, R. A., 1934. Statistical Methods for Research Workers. Oliver and Boyld Ltd.

4. Conclusion: Add more about the negativity of excessive coffee consumption in the last line so readers know about the importance of limiting coffee consumption in addition to the benefits that might arise. Add more about contraindication of coffee consumption (positivity and negativity) for each methods as a suggestion for readers which method of the coffee they will consume.

This reviewer suggestion was included on page 28 line 595-599 (file ‘Manuscript’).

Reviewer #2: 

1. The abstract is too long, please short it.

During the building of the manuscript in PDF format, there was an error, and the abstract was presented twice. This may have caused some confusion about the length of the abstract as it contains 210 words. Anyway, the abstract was shortened, especially in the introduction and the methods were removed. 

2. Most of the data show that FCS was not effective in controlling obesity. I doubt that the consumption of FCS may have no positive health effect in rats fed with a high-fat diet.

The consumption of FCS promoted some positive health effects as described in the manuscript. FCS consumption increased Bifidobacterium populations and HDL-c reverse cholesterol transport to tissues and reduced Il1b mRNA levels. 

The lack of a significant effect of coffee on body weight gain of rats fed with a high-fat diet seems to be explained by the reduced food intake presented by the high-fat diet rats during the 8-week treatment period, associated with the short period of treatment (8 weeks) and the studied sample composed by newly weaned rats. These variables seem to have attenuated weight gain and dyslipidemia and may justify the lack of some positive health effects of coffee. The growth curve of the rats (figure presented in the file 'Response to Reviewers') clearly shows that only after 3 weeks of treatment high-fat fed rats begin to present a significantly higher body weight gain than control rats. Also is possible to note that only after 7 weeks of treatment the growth curve of the HF (+) group no longer appears superimposed to that of HF (-) group, suggesting that a longer time of treatment could exacerbate the obesity state and consequently the possible positive effect of coffee on body weight.

The literature shows that the treatment with coffee for periods longer than 8 weeks [Cowan et al 2014, Vitaglione et al, 2019] and samples composed of adult animals instead of newly weaned animals [Ilmiawati et al 2020, Rustandi et al, 2019] seem to promote a more pronounced obesity state and a consequently higher impact of coffee on body weight and lipid profile.

Cowan, T. E., Palmnäs, M. S., Yang, J., Bomhof, M. R., Ardell, K. L., Reimer, R. A., Vogel, H. J., Shearer, J. Chronic coffee consumption in the diet-induced obese rat: impact on gut microbiota and serum metabolomics. The Journal of nutritional biochemistry. 2014; v. 25, n. 4, p. 489–495.

Vitaglione P, Mazzone G, Lembo V, D'Argenio G, Rossi A, Guido M, Savoia M, Salomone F, Mennella I, De Filippis F, Ercolini D, Caporaso N, Morisco F. Coffee prevents fatty liver disease induced by a high-fat diet by modulating pathways of the gut-liver axis. J Nutr Sci. 2019 Apr 22;8:e15. doi: 10.1017/jns.2019.10. PMID: 31037218; PMCID: PMC6477661.

Ilmiawati C, Fitri F, Rofinda ZD, Reza M. Green coffee extract modifies body weight, serum lipids, and TNF-α in high-fat diet-induced obese rats. BMC Res Notes 2020;13. https://doi.org/10.1186/s13104-020-05052-y

Rustandi F, Aman IGM, Pinatih GNI. Administration of bali arabica (Coffea arabica) coffee extracts decreases abdominal fat and body weight in obese Wistar rats (Rattus norvegicus). Indones J Anti-Aging Med 2019;3

Figure (in the file 'Response to Reviewers'): Body weight of rats treated with control (CT) or high-fat (HF) diet added (+) or not (-) of coffee. CT (-) control diet AIN-93G; CT (+) control diet + coffee; HF (-) high-fat diet; HF (+) high-fat diet + coffee. Values are means ± S.E., n = 7/group. * P < 0.05 for control vs. high-fat diet.

Reviewer #3: 

1. The language of the manuscript is not very good and needs to be strengthened. 

The language of the manuscript was revised by a native English speaker specialized in editing research manuscripts. The two certificate of English editing (first and second reviews) is attached in the file 'Response to Reviewers'.

2. References must be refreshed, some of them are too old. 

We try to refresh the references. The older references, such as 29 and 40, refer to traditional methodologies that have not been updated so far.

3. The sequence number of the figure is confused. 

The sequence number of the figure was corrected in the text. 

5. How to conduct dietary intervention for animals? How to determine that the provided feed has been completely consumed by the animal? 

Daily, in the afternoon, the amount of feed to be provided was weighed and in the morning of the next day, feed rest was weighed. The difference between the amount of feed provided and the amount of feed rest was considered as feed intake.

This data was described in the text on page 6, line 148-149 (file ‘Manuscript’).

7. How to determine the amount of freeze-dried coffee added? Why choose 3.9 g?

The requested explanations were included in the manuscript in the section material and methods subitem “2.1. Preparation of freeze-dried coffee solution” on page 5, line 119-128 (file ‘Manuscript’).

The amount of coffee solution added to rats’ diet was defined considering the estimated average usual daily coffee intake of 163mL and the usual food amount of 1,290g consumed by the Brazilian population. Considering a mean daily amount of food consumed by an adult rat of 25g, the equivalent dose estimated for rats’ average coffee intake would be 3.15mL/day, resulting in a 126mL coffee/kg diet. After the freeze-drying process, 126mL of 10% coffee solution yielded 3.9g of powder. The freeze-dried coffee solution was mixed with the other diet components until obtaining a homogenous mixture, and subsequently hydrated and pelleted.

---

## [Decision Letter · Decision Letter 1]

1 Dec 2021

PONE-D-21-14462R1Effect of a freeze-dried coffee solution in a high-fat diet-induced obesity model in rats: biochemical and inflammatory impacts and effects on gut microbioPLOS ONE

Dear Dr. Cavalcanti,

Thank you for submitting your manuscript to PLOS ONE. After careful consideration, we feel that it has merit but does not fully meet PLOS ONE’s publication criteria as it currently stands. Therefore, we invite you to submit a revised version of the manuscript that addresses the points raised during the review process.

 The revised manuscript showed significant improvement, however, there are still several issues that need to be addressed.  Please see specific comments below. In addition, the quality of the language needs to be improved.  We suggest you thoroughly copyedit your manuscript for language usage, spelling, and grammar. If you do not know anyone who can help you do this, you may wish to consider employing a professional scientific editing service.

Specific comments:

1. Line 38, replace "Independently" with "Regardless".

2. Line 126, change "*Coffea arabica*" to "*C. arabica*".

3. Line 127 - 128, there are 4 numbers here with only 3 caffeoylquinic acid listed. Please clarify.

4. Line 132, replace "n°" with "no."

5. Line 176 - 177, change "San Luis" to "St. Louis".

6. Line 199, This is confusing.  Is 10μl the total reaction volume? Please rephrase for clarity.

7. Line 211 - 214, please follow scientific nomenclature rules and use abbreviation here since these bacteria were mentioned earlier. For example:  *E. coli, E. faecalis, B. lactis*, &* L. plantarum.*

8. Line 407, change "independently" to "independent"

9. Line 445, change “*Coffeea*” to “*C.*”

10. Line 530, change "pathogen bacteria" to pathogenic bacteria"

11. Line 558 - 559, replace “*Bifidobacterium*” to “*B.*”

12. Line 597 - 599, the statement "..., while some health..." was not supported by the data presented, please delete.

We look forward to receiving your revised manuscript.

Kind regards,

Baochuan Lin, Ph.D.

Academic Editor

PLOS ONE

Journal Requirements:

Reviewers' comments:

Reviewer's Responses to Questions

**Comments to the Author**

1. If the authors have adequately addressed your comments raised in a previous round of review and you feel that this manuscript is now acceptable for publication, you may indicate that here to bypass the “Comments to the Author” section, enter your conflict of interest statement in the “Confidential to Editor” section, and submit your "Accept" recommendation.

Reviewer #4: (No Response)

2. Is the manuscript technically sound, and do the data support the conclusions?

Reviewer #4: Yes

3. Has the statistical analysis been performed appropriately and rigorously? 

Reviewer #4: Yes

4. Have the authors made all data underlying the findings in their manuscript fully available?

Reviewer #4: Yes

5. Is the manuscript presented in an intelligible fashion and written in standard English?

Reviewer #4: Yes

6. Review Comments to the Author

Reviewer #4: The authors have determined the effect of FCS on on the physiological parameters, lipid profile, and microbiota in the HF-diet-fed rats. For this version, the comments are addressed carefully and is recommended for publication.

7. PLOS authors have the option to publish the peer review history of their article (what does this mean?). If published, this will include your full peer review and any attached files.

Reviewer #4: No

---

## [Author Response · Author response to Decision Letter 1]

20 Dec 2021

Dear PhD Baochuan Lin,

Thank you by the opportunity to submit a revised version of our manuscript “Effect of freeze-dried coffee solution in a high-fat diet-induced obesity model in rats: biochemical and inflammatory impacts and effects on gut microbiota”. Below are the responses to each specific comment brought up. All changes made were marked on the revised manuscript copy, and the answers to editor questions are presented below. According to editor suggestion, the manuscript was copy edited for language usage, spelling, and grammar by Editage (editage.com.br). In the file "Response to Reviewers" is the Editing Certificate issued by Editage.

Yours sincerely,

Marilia Cavalcanti, M.Sc.

Corresponding author

Postgraduate Program in Human Nutrition, Faculty of Health Sciences, Campus Universitário Darcy Ribeiro, Universidade de Brasília, Brasília, POBox 70910-900, Brazil. Phone +(55) - 61 - 31070092 e-mail: mariliaunb@outlook.com

Editor specific comments:

Specific comments:

1. Line 38, replace "Independently" with "Regardless". Accepted.

2. Line 126, change "Coffea arabica" to "C. arabica". Accepted.

3. Line 127 - 128, there are 4 numbers here with only 3 caffeoylquinic acid listed. Please clarify. It was revised.

4. Line 132, replace "n°" with "no." It was revised.

5. Line 176 - 177, change "San Luis" to "St. Louis". Accepted.

6. Line 199, This is confusing. Is 10μl the total reaction volume? Please rephrase for clarity. It was revised.

7. Line 211 - 214, please follow scientific nomenclature rules and use abbreviation here since these bacteria were mentioned earlier. For example: E. coli, E. faecalis, B. lactis, & L. plantarum. It was revised.

8. Line 407, change "independently" to "independent" It was revised.

9. Line 445, change “Coffeea” to “C.” Accepted.

10. Line 530, change "pathogen bacteria" to pathogenic bacteria" It was revised.

11. Line 558 - 559, replace “Bifidobacterium” to “B.” Accepted.

12. Line 597 - 599, the statement "..., while some health..." was not supported by the data presented, please delete. It was deleted.

---

## [Editor Report · Decision Letter 2]

21 Dec 2021

Effect of a freeze-dried coffee solution in a high-fat diet-induced obesity model in rats: impact on inflammatory response, lipid profile, and gut microbiota

PONE-D-21-14462R2

Dear Dr. Cavalcanti,

We’re pleased to inform you that your manuscript has been judged scientifically suitable for publication and will be formally accepted for publication once it meets all outstanding technical requirements.

Kind regards,

Baochuan Lin, Ph.D.

Academic Editor

PLOS ONE
---

## [Editor Report · Acceptance letter]

17 Jan 2022

PONE-D-21-14462R2 

Effect of a freeze-dried coffee solution in a high-fat diet-induced obesity model in rats: impact on inflammatory response, lipid profile, and gut microbiota 

Dear Dr. Cavalcanti:

I'm pleased to inform you that your manuscript has been deemed suitable for publication in PLOS ONE. Congratulations! Your manuscript is now with our production department. 

Kind regards, 

on behalf of

Dr. Baochuan Lin 

Academic Editor

PLOS ONE